



# Correlation between the fractal of aftershock spatial distribution and active fault on Sumatra

Bahary Setyawan[1], Benyamin Sapiie[1]

[1]Department of Engineering Geology, Institut Teknologi Bandung, Bandung, 40132, Indonesia

*Correspondence to*: Bahary Setyawan (baharys.bs@gmail.com)

**Abstract.** This study discusses the correlation between the fractal of spatial epicentre distribution of aftershock ($D_2$) and active fault ($D_0$) in the Sumatra region. We identified 15 earthquakes in this region that were followed by aftershock cluster

and related to the Sumatra Fault Zone or Southern Andaman West Fault. The spatial epicentre distribution of the aftershock was estimated by using two-point correlation integral and the $D_2$ values found were varying from $1.03 \pm 0.03$ to $1.68 \pm 0.08$. We estimated the fractal dimension of the active fault by using Box-Counting Method and found that the variation of $D_0$ values in the range of $0.95 \pm 0.03$ to $1.16 \pm 0.01$. Positive correlation was found in this study and two patterns were identified that had similar slope with different intercept. However, there was also a correlation that had steeper slope. The

steeper slope was related to earthquake doublet mechanism that could generate more random spatial distribution of the aftershock in the fault system.

## 1 Introduction

Sumatra tectonic is an oblique convergence that generates slip segmentations. These segmentations are dip-slip in the fore arc (subduction zone) and strike slip in the arc which is Sumatra Fault Zone (SFZ) as dextral fault (McCaffrey et al., 2000).

SFZ is one of the most active faults in the world (Daryono, et al., 2016). Seismicity and active fault in Sumatra, especially fractal studies, has been carried out many times by other researcher. However, only a few researches have specifically studied about this active fault.

Earthquake itself is an event that has an undoubtedly complex pattern, but this can be quantified by using fractal concept

(Donald L. Turcotte, 1997). Fractal has been widely used to develop physical models of earthquake genesis including non-linear dynamic and it can be used to understand seismic pattern in a region (Dimri, 2005). Fractal also follow self-organized critical (SOC) which state that nature has its own order, when at the critical point it will become the same pattern. In seismology for instance, the Gutenberg-Richter equation regarding b-value is considered as SOC.





The fractal dimension of faults is likely related to the spatial distribution of earthquakes and strain release modes (Okubo and Aki, 1987). There is a positive correlation between the fractal of spatial distribution of aftershock and pre-existing active fault (Nanjo and Nagahama, 2004). In the same study the correlation between both fractal values is SOC, the equation being as follows:

$$D_2 = (0.72 \pm 0.02)\,D_0 + (0.91 \pm 0.02) \tag{1}$$

Where $D_2$ is a fractal the spatial distribution of epicentre aftershocks and $D_0$ is the active fractal fault dimension.

Yamashita and Knopoff (1987) found that the interaction between slipping main faults and the surrounding small satellite faults is more suitable for simulating aftershock occurrences. That result is based on assumption that the probability
distribution of critical distances for quasistatic and catastrophic crack growth has a power law distribution. Libicki and Ben-Zion (2005) discussed that the fractals of fault geometry cannot be easily inferred by an epicentre. Padhy et al., ( 2013) found no correlation between $D_2$ and $D_0$ in Sumatra based on their study in aftershock sequences from Aceh Earthquake 2004. This result is more affected by the large uncertainty relative to the variation of $D_2$.

It is intriguing to confirm whether Eq. 1 could hold for Sumatra as well. If the equation 1 is consistent, it can be used as a model to enhance the understanding of the aftershocks occurrences. The fractal term in particular self-organized critical or scale invariance may not be found in general region due to chaos and uncertainty in earthquake itself. Therefore, understanding relationship between pre-existing active fault and spatial distribution of aftershocks in fractal approach becomes our concern.

**2 Data and Methodology**

**2.1 Data**

The data were obtained from local station in PuSGeN catalogue (National Earthquake Study Centre) which was collected from International Seismological Centre (ISC), National Earthquake Information Centre U.S. Geological Survey (NEIC-USGS), catalogue that has relocated by Engdhal (Engdahl et. al., 1998) and relocated Badan Meteorologi, Klimatologi, dan
Geofisika (BMKG) catalogue (Shiddiqi et. al., 2015). This catalogue has all the earthquake data with the magnitude above 4.5 $M_w$, starting from 1900 to 2016. In this study, aftershocks are defined as earthquakes that occur after the main shock and have a smaller magnitude. The analysed aftershocks are the earthquakes that are not related to the Benioff Zone or subduction zone. In other words, the analysis was only performed on the earthquakes related to the Sumatra Fault Zone (SFZ) and West Andaman Fault (WAF). Shallow earthquake is limited to 50 km depth, considering the lower limit of the
seismogenic zone in Sumatra (Collings et al., 2012; Klingelhoefer et al., 2010).





The surface geometry of active faults was also obtained from PuSGeN. Active faults were identified based on morphology that had been carried out in previous studies. The SFZ was divided into 42 segments, while the WAF was divided into 4 segments. These active faults are defined as a fault that has moved less than 10,000 years. Tab. 1 is a tabulation of the results of the identified aftershocks and the correlation result, while Figure 1 is the distribution of the earthquake position in this

study. Aftershocks occur more in the North than in the South, especially on land.

**Table 1: Mainshock data responsible for the earthquake and the result of $D_2$ and $D_0$ estimation and correlation.**

| Earthquake | Location | | Date | Mag. (Mw) | Depth (km) | $D_0$ | $D_2$ | Correlation |
|---|---|---|---|---|---|---|---|---|
| | Long. | Lat. | | | | | | |
| Nicobar, 1996 | 94,357 | 6,956 | 12-Apr-96 | 5,8 | 23,4 | 1,06 (± 0,03) | 1,68 (± 0,08) | Correlation 1 Nanjo dan Nagahama (2004) |
| Aceh Jaya, 2013 | 95,956 | 4,794 | 21-Jan-13 | 6,1 | 9,58 | 1,09 (± 0,03) | 1,67 (± 0,13) | |
| Nicobar, 2014 | 94,231 | 7,593 | 21-Mar-14 | 6,4 | 24,8 | 0,95 (± 0,03) | 1,64 (± 0,03) | |
| Nicobar, 1967 | 93,586 | 8,648 | 02-Jul-67 | 6,1 | 6,5 | 1,14 (± 0,04) | 1,49 (± 0,11) | Correlation 2 |
| Sunda Strait,1985 | 105,123 | -6,031 | 10-Augst-85 | 5,6 | 22,3 | 1,09 (± 0,02) | 1,43 (± 0,09) | |
| Sunda Strait, 2002 | 105,205 | -6,314 | 15-Jan-02 | 6,1 | 10 | 1,07 (± 0,01) | 1,43 (± 0,04) | |
| Tanah Datar, 2007 | 100,498 | -0,493 | 06-Mar-07 | 6 | 19 | 1,06 (± 0,02) | 1,34 (± 0,04) | |
| Nicobar, 1970 | 93,848 | 9,068 | 25-Oct-70 | 6,4 | 24,6 | 1,03 (± 0,01) | 1,30 (± 0,07) | |
| East Aceh, 2003 | 97,568 | 4,495 | 22-Jan-03 | 5,8 | 33 | 0,99 (± 0,04) | 1,23 (± 0,05) | |
| Pidie, 1967 | 96,236 | 5,079 | 12-Apr-67 | 6,7 | 17,5 | 1,16 (± 0,01) | 1,14 (± 0,03) | Correlation 3 |
| Naganraya, 1997 | 96,405 | 4,284 | 20-Augst-97 | 6 | 9,5 | 1,14 (± 0,03) | 1,13 (± 0,03) | |
| Tapanuli, 2008 | 99,147 | 1,64 | 19-May-08 | 6 | 10 | 1,09 (± 0,02) | 1,07 (± 0,14) | |
| Mandailingnatal, 2006 | 99,859 | 0,626 | 17-Dec-06 | 5,8 | 30 | 1,05 (± 0,02) | 1,03 (± 0,03) | |
| Tapanuli, 2005 | 99,093 | 1,751 | 29-Mar-05 | 5,5 | 23,4 | 1,03 (± 0,01) | 1,20 (± 0,02) | *Outlier* |
| Nicobar, 2015 | 94,648 | 6,843 | 05-Nov-15 | 6,6 | 10 | 1,05 (± 0,02) | 1,15 (± 0,02) | |

**2.2 Methodology**

Fractal of spatial distribution of aftershock is estimated from two-point correlation integral that evaluate distance of each epicentre (e.g., Grassberger and Procaccia, 1983; Nanjo and Nagahama, 2004; Roy et al., 2011). The equation below:

$$C(r) = \frac{1}{N^2} \sum_{i \neq j} H(r - R) \tag{2}$$

Where C (r) is a two-point correlation and N is the number of aftershocks event. H is a Heaviside function where the value H = 1 if z ≥ 0 and H = 0 if z <0. r is a certain distance, while R is the distance between epicentres. To calculate the distance

between points (R), the equations of spherical triangles (Hirata, 1989) are used:

$$R = \cos^{-1} \{\cos \alpha_1 \cos \alpha_2 + \sin \alpha_1 \sin \alpha_2 \cos(\beta_1 - \beta_2)\} \tag{3}$$




Where α is latitude and β is longitude. Furthermore, the fractal dimension of aftershocks distribution is obtained by the power law relationship between C (r) and r, with the equation below:

$$C(r) \sim r^{D_2} \qquad (4)$$

$D_2$ is a fractal spatial distribution of aftershocks. The greater the D2 value, the distribution of epicentre points will be

5   increasingly random and has a maximum value of 2 . Conversely, a small D2 value indicates a tight earthquake distribution.

Fractal dimension of active faults was estimated by using the standard box-counting method, as in Donald L. Turcotte (1997). The box-counting method is to make boxes in the aftershock cluster. Then, the box passed by the fault will be calculated in number (N (r)) and the length of the box (r). Finally, the fractal value is obtained from the power law between

10  N (r) and r with the negative correlation relationship, as follows:

$$N(r) \sim r^{-D_0} \qquad (5)$$

Where $D_0$ is the active fault fractal dimension, the greater the value shows more irregular shape.

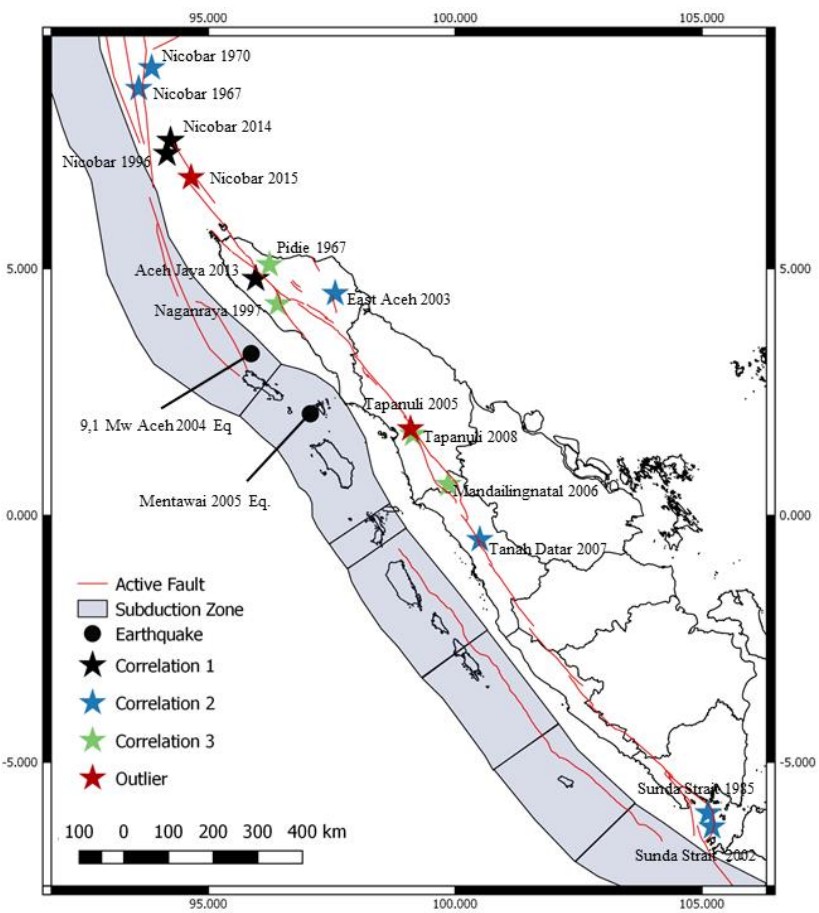

Figure 1: spatial distribution of earthquake events





# 3 Results

## 3.1 Fractal of spatial distribution of aftershock ($D_2$) and active fault ($D_0$)

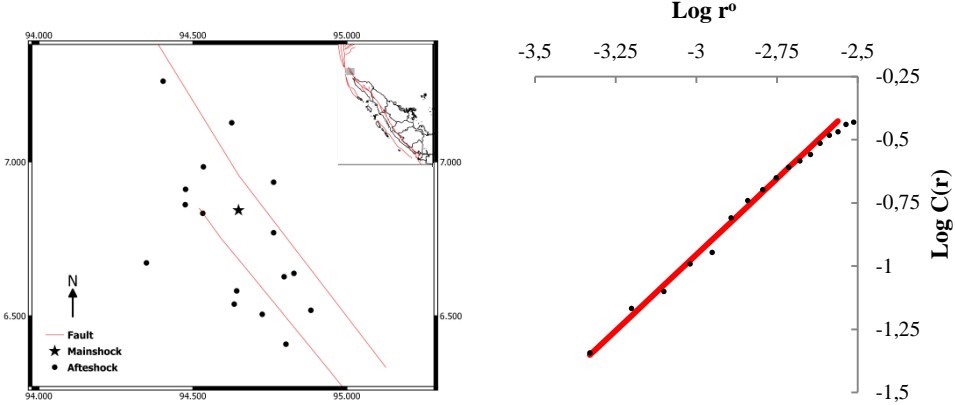

**Figure 2: Spatial distribution of aftershock of Nicobar 2015 earthquake (left) and double logarithmic curve between r and C(r) (right).**

Fig. 2 is the distribution of aftershocks that occurred in the 2015 Nicobar earthquake and also the double logarithmic curve to estimate the $D_2$ value. The C(r) value calculation was conducted by using Python 3.7. $D_2$ estimation has a fairly high level of biases due a lack of complete catalogue or limited number of data points. (Padhy et al., 2013). The estimations were performed multiple times in several points to find a lower error value as done by Nanjo & Nagahama (2004). At points 1 to 15, the gradient value on the logarithm curve was 1.16 with error 0.025. Then the value with points 1 to 16 was 1.14 with an error of 0.023 while at points 1 to 17 had a value of $D_2$ 1.14 and error was 0.024. The value taken in this $D_2$ estimation was 1.15. Similar algorithm also used in estimating the fractal of spatial distribution of other aftershocks.

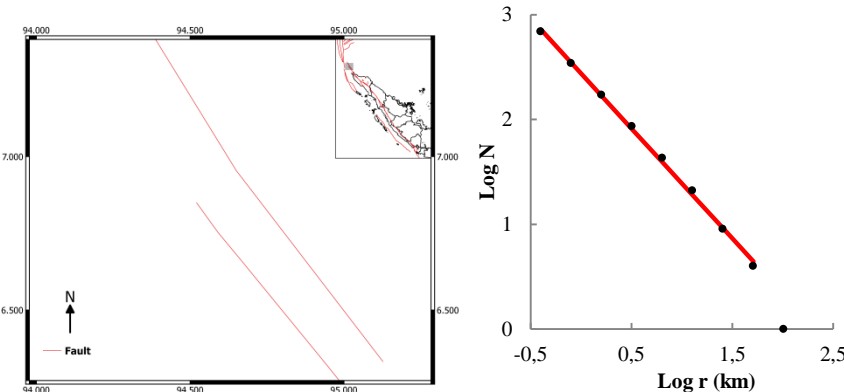

**Figure 3: Northern Sumatra Fault Zone (SFZ), Nicobar & Seulimeum N Segment (left) and double logarithmic curve between r and N (right).**





The fault that was responsible for Nicobar 2015 earthquake is SFZ, which is in Nicobar and Seulimeum N Segment (Fig. 3). Box-Counting method was performed on these two segments where aftershocks occurred. The estimation of $D_0$ was also taken from several points which eventually the lowest error was chosen. At points 2 to 7 the slope was 1.08 with an error of 0.024. Then, the values of 2 to 8 resulted in fractal dimension values of 1.05 and error of 0.019. In other estimation, at point 2 to 9, the value of 1.05 and error 0.016 is obtained. Then the $D_0$ value with the lowest error of 1.05 was chosen. The other fractal dimensions of active fault were also carried out in the same way. The calculation results of the $D_2$ and $D_0$ values can be seen in Tab. 1.

### 3.2 Correlation between $D_2$ and $D_0$

Correlation between $D_2$ and $D_0$ were done using least-squared regression. We found three correlations and two earthquake occurrences that did not follow the three correlations (Tab. 1 and Fig. 4). First correlation was indicated by the $D_2$ value of more than 1.6, similar with Nanjo and Nagahama (2004) correlation. For the second correlation, it was found that the gradient was steeper than the other correlation. The equation of correlation 2 is as follows:

$$D_2 = (1.88 \pm 0.02)D_0 - (0.63 \pm 0.23) \tag{6}$$

This correlation had a $D_2$ value ranging from 1.2 to 1.5. Last correlation was slightly the same as equation 1, but had a significant intercept difference. The equation is as follow:

$$D_2 = (0.95 \pm 0.15)D_0 + (0.04 \pm 0.17) \tag{7}$$

The two earthquake events that did not follow the three correlations above are the Tapanuli 2005 and Nicobar 2015 earthquakes.

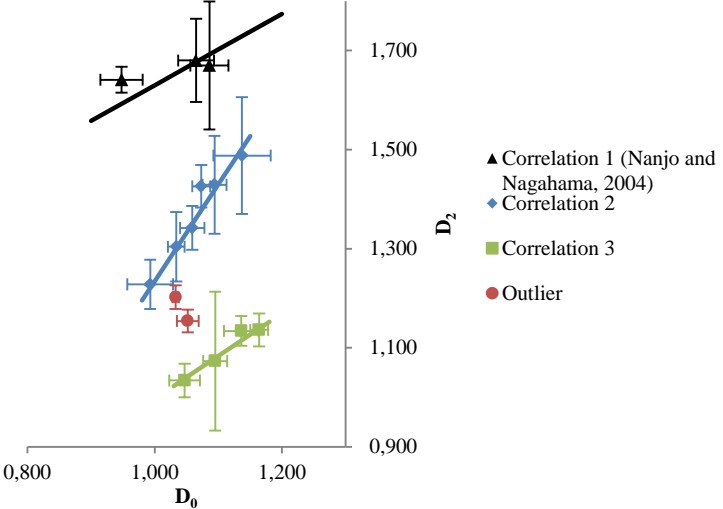

**Figure 4: Correlation between $D_2$ and $D_0$. There are 3 patterns which first pattern follows Nanjo and Nagahama (2004) correlation. The Second correlation shows steeper gradient than the others. The last correlation is slightly similar with correlation 1, but had a difference in intercept.**



## 4 Discussions

The estimation of the $D_2$ value or fractal of earthquake distribution in Sumatra region has been conducted by previous researchers such as Padhy et al., (2013), Roy et al., (2011), and Bhattacharya and Kayal, (2007). The studies show different results due to different catalogues used. In addition, Roy et al. (2011) carried out their study using all epicentre data including shallow to deep earthquakes (Benioff Zone). Padhy et al. (2013) and Bhattacharya and Kayal (2007) analysed the spatial distribution of aftershocks in 9.1 Mw Aceh 2004 earthquake. However, that study was outside of our scope. Sukmono et al. (1996) estimated fractal dimensions of active fault on land part of Sumatra or in SFZ. The results obtained are relatively similar, but there is a significance difference in Oreng and Aceh-Center segment. It is due to differences in the morphological interpretation of active faults in that segments.

The three earthquake events in correlation 1, namely the Nicobar 1996, Aceh Jaya 2013 and Nicobar 2014 earthquake, show similar results as Nanjo and Nagahama (2004). Although it has a difference in the earthquake magnitude threshold, this result is consistent with the study. Nanjo and Nagahama (2004) used earthquake data above 0 Mw. The third and the first correlation show a nearly identical value with only 0.06 differences. A significant difference in the intercept shows the scale variation. It seems that the magnitude of earthquake event or aftershocks do not affect the difference in the intercept. We estimated b-value variation for both correlation 1 and 3 to find out whether there was an influence of the value (Tab. 2). From the estimations of b-value, there were significant differences. However, the estimations had quite high uncertainty level in some earthquake event. Different catalogue also did not show significant differences. The number of the aftershock looked quite significant. In correlation 1, the number of aftershocks was above 18, whereas in the third the opposite occurred. However this result also cannot strongly explained why there was a scale difference in the number of aftershocks generated.

Table 2: Comparison of b-value results with previous studies (Roy, et al., 2010), catalogue of PuSGeN with a threshold of 4.5 Mw and USGS catalogue with a threshold of 2.5 Mw. The b-value estimation was conducted by the least-squared fit method.

| Earthquake | B-Value (Roy, et al., 2010) | B-Value (PuSGeN Catalogue) | B-Value (USGS Catalogue) | Correlation |
|---|---|---|---|---|
| Nicobar, 1996 | 1,65-1,75 | 1,90* | 0,485* | Correlation 1 |
| Aceh Jaya, 2013 | 1,35-1,45 | 1,96 | 0,900 | (Nanjo and |
| Nicobar, 2014 | 1,65-1,75 | 2,03 | 1,353 | Nagahama, 2004) |
| Pidie, 1967 | 1,35-1,45 | 1,05 | - | |
| Naganraya, 1997 | 1,35-1,45 | 0,80* | 1,042 | Correlation 3 |
| Tapanuli, 2008 | 1,65-1,75 | 0,95* | 1,076 | |
| Mandailingnatal, 2006 | 1,65-1,75 | 0,83* | 1,681 | |

* Estimation results have high uncertainty





The correlation 2 shows a significant difference in the slope gradient. One of the earthquake events in this correlation was the Tanah Datar 2007, which occurred in Sumani Segment and show the earthquake doublet evidence (Daryono, 2016). Doublet earthquake is identified by a magnitude difference of no more than 0.2, a distance of no more than 100 km, with
time distance of no more than one year (Astiz and Kanamori, 1984; Lay and Kanamori, 1980), while other study stated that magnitude difference could be less than 0.4 (Felzer et al., 2004). However, mechanism of this earthquake is not completely defined (Gibowicz et al., 2003). Tab. 4 shows a comparison between the main shock and the largest foreshock or aftershock to identify the possibility of another earthquake doublet. There were only three earthquakes included in the criteria, namely Nicobar 1967, Sunda Strait 2002 and Tanah Datar 2007 earthquake. The other earthquakes had magnitude difference of less
than 4, but had time difference more than one year. If we assume that all events in correlation 2 were doublet earthquakes, then the mechanism for the occurrence of aftershocks was likely to cause a difference in slope. The doublet earthquake event mechanism in main shock and aftershocks were the same and the rate of aftershock was higher (Felzer et al., 2004).

**Table 3: Comparison of magnitude between main shock and highest magnitude in foreshock or aftershock. Correlation 2 shows**
**difference in magnitude less than 0.4.**

| Earthquake | Mainshock | | | | Highest Mag. foreshock or aftershock | | | | Mag. Diff. (Mw) | Correl ation |
|---|---|---|---|---|---|---|---|---|---|---|
| | Loc. | | Date | Mag. (Mw) | Loc. | | Date | Mag. (Mw) | | |
| | Long. | Lat. | | | Long. | Lat. | | | | |
| Nicobar, 1996 | 94,357 | 6,956 | 12/04/1996 | 5,8 | 94,294 | 7,032 | 13/04/1996 | 5,4 | 0,4 | Corr. 1 |
| Aceh Jaya, 2013 | 95,956 | 4,794 | 21/01/2013 | 6,1 | 95,838 | 4,986 | 22/10/2013 | 5,4 | 0,7 | |
| Nicobar, 2014 | 94,231 | 7,593 | 21/03/2014 | 6,4 | 94,36 | 7,452 | 18/11/2014 | 5,6 | 0,8 | |
| **Nicobar, 1967** | **93,586** | **8,648** | **02/07/1967** | **6,1** | **93,671** | **8,704** | **19/07/1967** | **5,8** | **0,3** | Corr. 2 |
| **Sunda Strait,1985\*** | **105,123** | **-6,031** | **10/08/1985** | **5,6** | **105,248** | **-6,588** | **15/03/1984** | **5,5** | **0,1** | |
| **Sunda Strait, 2002** | **105,205** | **-6,314** | **15/01/2002** | **6,1** | **105,205** | **-6,314** | **15/01/2002** | **6,1** | **0** | |
| **Tanah Datar, 2007** | **100,498** | **-0,493** | **06/03/2007** | **6,4** | **100,53** | **-0,488** | **06/03/2007** | **6,3** | **0,1** | |
| **Nicobar, 1970\*** | **93,848** | **9,068** | **25/10/1970** | **6,4** | **92,924** | **10,122** | **05/11/1971** | **6,04** | **0,36** | |
| **East Aceh , 2003\*** | **97,568** | **4,495** | **22/01/2003** | **5,8** | **97,81** | **4,23** | **22/12/2004** | **5,6** | **0,2** | |
| Pidie, 1967 | 99,093 | 1,751 | 12/04/1967 | 6,7 | 96,333 | 5,273 | 12/04/1967 | 5,6 | 1,1 | Corr. 3 |
| Naganraya, 1997 | 94,648 | 6,843 | 20/08/1997 | 6 | 95,759 | 4,929 | 15/02/1999 | 5,5 | 0,5 | |
| Tapanuli, 2008 | 96,236 | 5,079 | 19/05/2008 | 6 | 99,18 | 1,724 | 19/05/2008 | 5,3 | 0,7 | |
| Mandailingnatal 2006 | 96,405 | 4,284 | 17/12/2006 | 5,8 | 99,96 | 0,5 | 17/12/2006 | 5,4 | 0,4 | |
| Tapanuli, 2005 | 99,147 | 1,640 | 29/03/2005 | 5,5 | 98,92 | 2,05 | 29/03/2005 | 5,3 | 0,2 | Outlier |
| Nicobar, 2015 | 99,859 | 0,626 | 08/11/2015 | 6,6 | 94,534 | 6,984 | 08/11/2015 | 5,6 | 1 | |

\* The earthquake that show more than one year in difference time.





In other earthquake events, Tapanuli 2005 and Nicobar 2015, there were no correlations between the spatial distributions of aftershocks with active faults. This indicates that the mechanism of different aftershocks in these two earthquakes did not occur in other earthquakes. The aftershocks might be generated due to the development of satellite fractures that do not correlate with the main fault (Yamashita and Knopoff, 1987). The other alternatives are these two earthquakes were not

earthquake clusters caused by active pre-existing faults. However, an earthquake was triggered due to a higher earthquake around it or aftershocks from other earthquakes. A year before Nicobar 2015 earthquake, a preceding earthquake occurred in the north (Nicobar 2014 earthquake), which location was less than 100 km from Nicobar 2015 location. Then in the Tapanuli 2005 earthquake, there was a 9.1 Aceh 2004 earthquake a few months earlier and then Mentawai 2005 in the same date with Tapanuli 2005 (Fig. 1).

**5 Conclusion**

A positive correlation was found between the fractal distribution of aftershocks and active faults. There was self-organized critical and scale invariance between spatial distribution of aftershock and active fault. The mechanism of the earthquake or aftershocks will affect how the correlation between aftershocks spatial distribution and pre-existing active faults. Significance difference in intercept between correlation 1 and 3 was likely to be the variation of b-value or number of the

aftershock. Finally, this study can be used as a model to predict the spatial distribution of aftershocks with variations of general earthquakes, doublet earthquakes and scales.

**Acknowledgement**

We would like to acknowledge Indonesia Endowment Fund for Education (LPDP Indonesia) to fully support this research. We also would like to acknowledge National of Earthquake Study Center (PuSGeN) to grant a permission to use earthquake

catalogue and active fault map.

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
