# Peer review of "Correlation between the fractal of aftershock spatial distribution and active fault on Sumatra"

_Natural Hazards and Earth System Sciences, 2019_

## Referee Comment (RC1) · Francesco Visini (Referee) · 27 Oct 2019

Dear Editor, I have now completed my review of the manuscript "Correlation between the fractal of aftershock spatial distribution and active fault on Sumatra" by Bahary Setyawan and Benyamin Sapiie. The methodologies adopted have a precedent in the scientific literature and the data can be assumed to be of sufficient quality for such an analysis however, I would like to suggest some notes to the Authors to improve the quality of the manuscript. Very important, moreover, is a necessary review of the quality of writing to achieve a level that is acceptable for publication. I furnish a pdf file with an evaluation following the NHESS standard and guidelines for reviewing,

and some points that I would like to suggest to the Authors to address in the revised version. Sincerely, Francesco Visini

Please also note the supplement to this comment:
https://www.nat-hazards-earth-syst-sci-discuss.net/nhess-2019-215/nhess-2019-215-RC1-supplement.pdf

[Figure]

**Supplement:**

Dear Editor,

I have now completed my review of the manuscript "Correlation between the fractal of aftershock spatial distribution and active fault on Sumatra" by Bahary Setyawan and Benyamin Sapiie.

The methodologies adopted have a precedent in the scientific literature and the data can be assumed to be of sufficient quality for such an analysis however, I would like to suggest some notes to the Authors to improve the quality of the manuscript. Very important, moreover, is a necessary review of the quality of writing to achieve a level that is acceptable for publication.

I give an evaluation following the NHESS standard and guidelines for reviewing, and some points that I would like to suggest to the Authors to address in the revised version.

1) Does the paper address relevant scientific and/or technical questions within the scope of NHESS? Yes.

2) Does the paper present new data and/or novel concepts, ideas, tools, methods or results? New results.

3) Are these up to international standards? Yes.

4) Are the scientific methods and assumptions valid and outlined clearly? Scientific methods and assumptions are valid but not sufficiently clearly outlined throughout the manuscript.

5) Are the results sufficient to support the interpretations and the conclusions? The last point in the conclusions need to be explored in the discussions.

6) Does the author reach substantial conclusions? Yes

7) Is the description of the data used, the methods used, the experiments and calculations made, and the results obtained sufficiently complete and accurate to allow their reproduction by fellow scientists (traceability of results)? There are data and formula to reproduce calculations, but I think that the description of the data and some parts of the methodological section need to be rewritten to be sufficiently accurate.

8) Does the title clearly and unambiguously reflect the contents of the paper? Yes

9) Does the abstract provide a concise, complete and unambiguous summary of the work done and the results obtained? Yes.

10) Are the title and the abstract pertinent, and easy to understand to a wide and diversified audience? An explanation of fractal dimensions in the abstract is necessary to make it easy to understand to a wide and diversified audience.

11) Are mathematical formulae, symbols, abbreviations and units correctly defined and used? If the formulae, symbols or abbreviations are numerous, are there tables or appendixes listing them? Symbols need to be check.

12) Is the size, quality and readability of each figure adequate to the type and quantity of data presented? Yes.

13) Does the author give proper credit to previous and/or related work, and does he/she indicate clearly his/her own contribution? They cited the previous works, but I cannot see their individual contribution, I supposed to be equally distributed.

14) Are the number and quality of the references appropriate? References to support some statements need to be added.

15) Are the references accessible by fellow scientists? Yes

16) Is the overall presentation well structured, clear and easy to understand by a wide and general audience? No, the quality of the English, unfortunately, obscures many sentences.

17) Is the length of the paper adequate, too long or too short? I think it is adequate.

18) Is there any part of the paper (title, abstract, main text, formulae, symbols, figures and their captions, tables, list of references, appendixes) that needs to be clarified, reduced, added, combined, or eliminated? Appendix, which I think to be useful, need to be introduced in the manuscript, moreover explanations of what are figures and codes have to be given in the appendix itself.

19) Is the technical language precise and understandable by fellow scientists? Yes.

20) Is the English language of good quality, fluent, simple and easy to read and understand by a wide and diversified audience? No.

21) Is the amount and quality of supplementary material (if any) appropriate? Yes, but please introduce text to explain what is shown.

*Detailed comments.*

23) **Abstract**: Please, introduce a brief explanation of fractal dimensions to make it easy to understand to a wide and diversified audience.

**Page 1**

24) Line 18: Please specify what do you mean with "slip segmentation", it is not clear.

25) Line 20: I'd suggest to introduce here the figure1 with faults and seismicity to show the tectonics. Please, add also a wide tectonic scheme to introduce readers that are not-familiar with the region.

26) Line 21-22: Please, add the references to support this.

27) Line 25: the reference is Turcotte, 1997 (delete the name)

28) Line 28: Please, add the references to support that GR is considered as SOC.

**Page 2**

29) Line 4: I think it will be more clear than now if you remove ". In the same study the correlation between both fractal values is SOC, the equation being as follows" and just left ":".

30) Line 8: please specify what do you mean with "satellite faults".

31) Line 13: "This result is more affected by the large uncertainty relative to the variation of $D_2$.".Question1:  More than what? Question2: is there a reference to support this statement? Or, if it is your idea, it has to be supported by a quantitative analysis.

32) Line 22-25: Please rephrase because it is not clear.

33) Line 25: "This catalogue has all the earthquake data with the magnitude above 4.5 $M_w$, starting from 1900 to 2016", Please add a reference or show a completeness analysis to support this statement.

**Page 3**

34) Line 2: "Active faults were identified based on morphology that had been carried out in previous studies." Please add a reference

35) Line 3: "The SFZ was divided into 42 segments, while the WAF was divided into 4 segments" Who did this segmentation? and what it is based on?

36) Line 4: Are there paleoseismological studies to support the activity of these faults?

37) Line 5: can you be sure this is not a bias due to the seismic network configuration? How did you select the aftershock? Is a time-radius magnitude dependent approach?

**Page 4**

38) Regarding the mainshocks shown in the Table 1. You stated that "This catalogue has all the earthquake data with the magnitude above 4.5 $M_w$, starting from 1900 to 2016", but there are only earthquakes after the 1967 (I suppose this is the beginning of the instrumental period). So, this table is a selection of events? And if so, please specify the reasons. Or these are just the earthquakes with a correlation? And in this case you should show all the results.

39) At line 3 you wrote : "Tab. 1 is a tabulation of the results of the identified aftershocks and the correlation result". It is not clear if Tab 1 list the mainshocks. Please, specify how you build this table.

40) Finally, please use the dot for decimals.

41) Eq2 and later: Please use the same letter for distance R or r. Or are them different?

42) Line 11-13: Please rephrase because, I think a verb is missing. It is not clear here, how D2 was estimated. Please consider to explain variables in the order they are presented.

43) Line 6: "Conversely, a small D2 value indicates a tight earthquake distribution.", please specify what do you mean with a tight earthquake distribution. It is not clear if you are referring to the epicentral distribution or something else.

44) Line 6:  delete "Donald L. "

45) Line 7: The box-counting method is to make boxes in the aftershock cluster , please rephrase, because it is not clear.

46) Line 9: "length of the box (r)". This is the first time you mentioned length of box. Define all the parameters you are using to do your calculations.

**Page 5**

47) Line 6: please correct in: "Figure 2 shows". Specify that this is an example.

48) The C(r) value calculation was conducted by using Python 3.7  Python is a language, what do you mean? Is there a specific function that you used?

49) $D_2$ estimation has a fairly high level of biases due a lack of complete catalogue or limited number of data points. (Padhy et al., 2013). Can you give a quantitative analysis of this?

50) Line 8-9: "The estimations were performed multiple times in several points to find a lower error value as done by Nanjo & Nagahama (2004)", please specify lower than what.

51) Here you are introducing "points" for the first time. What are them? They are numbered, but you do not specify what is the range. Please specify what these points are and, if possible, add them in figures.

**Page 6**

52) Line 10: "We found three correlations and two earthquake occurrences that did not follow the three correlations (Tab. 1 and Fig. 4)", do you mean that "Basing on values of slopes and intercepts of Do versus D2, we identified 3 groups of D2-D0. For the first group, we calculated a correlation with a slope greater or equal than 1.6 and an intercept of X.X. For the second group, we calculated a correlation with a slope greater or equal than xxx and an intercept of X.X.....?  Please, rephrase this sentence because it is not very clear.

**Page 7**

53) Line 7: However, that study was outside of our scope. What do you mean?

54) Line 8: "The results obtained are relatively similar, but there is a significance difference in Oreng and Aceh-Center segment. It is due to differences in the morphological interpretation of active faults in that segments. Please, discuss in detail what are the reasons for such a differences, and how you define if they are significant or not.

55) Line 13: "The third and the first correlation show a nearly identical value with only 0.06 differences". Please specify which value, I think it is the slope, but you should be clear.

56) Line 13: " A significant difference in the intercept shows the scale  variation". What do you mean with scale variation?

57) Line 15: We estimated b-value variation for both correlation 1 and 3 to find out whether there was an influence of the value (Tab. 2).  How do you calculate the b? in table 2 you show b value much higher than 1-1.2, this is quite strange and needs to be explained in detail. Is that the b- of the sequence? Please specify in detail your computation

58) Line 18: "Different catalogue also did not show significant differences." For example, b-value moves from to 2 to 1.3, is that not significant?

59) Line 18-21: The number of the aftershock looked quite significant. In correlation 1, the number of aftershocks was above 18, whereas in the third the opposite occurred. However

this result also cannot strongly explained why there was a scale difference in the number of aftershocks generated. Please rephrase this sentence because it is not clear. What do you mean with "opposite" and "aftershocks generated" ?

**Page 8**

60) Line 10: correct in 0.4

61) I'd suggest to add a column in the table with the difference in the occurrence between mainshock and largest aftershock (in years).

**Page 9**

62) Line 5-9. This paragraph should be completely re-written, it is not clear.

63) In the discussions, there are no clues about the possible effect of completeness of the fault database and earthquake catalog on estimations of D2 ad Do and their correlation. Do the correlations show a spatial significance or not?

64) The last conclusion, "Finally, this study can be used as a model to predict the spatial distribution of aftershocks with variations of general earthquakes, doublet earthquakes and scales", it is not discussed in the manuscript before. If number of aftershock impact the b-value, and the correlations are done on basing on the observed number of aftershock, how is it possible predict spatial distributions? What do you mean with general earthquakes? Scales? Please, discuss this point in the discussion before state it in the conclusion.

Sincerely,
Francesco Visini

---

## Referee Comment (RC2) · Anonymous Referee #2 · 29 Oct 2019

This paper analyses the aftershock spatial distribution of 15 mainshocks, and the corresponding originating faults, in the Sumatra region, in terms of their correlation dimension and box-counting dimension, respectively. Then the authors find three classes of correlation (plus two cases of outliers). The ambitious intention of the authors is from this analysis to understand something about the earthquake mechanism produced by the corresponding fault system.

My surprise is the naïve application of the fractal methods to the available data. I am sorry to say, that the authors ignore most of the literature on the critical topics, and all related pitfalls occurring when applying blindly the fractal methods (in this case,

correlation dimension and box-counting). About this aspect, I would like to mention, just to remind one of the most important, Thelier (1990), from which it is clear that in the present paper there are many problems, for which I list only the main ones:

1. Too few data (especially in the aftershock spatial distribution)

2. Too small estimation of errors, because simply deduced from the linear regression in the log-log plot. If we estimate a more realistic error with an average of around three times that given, it is evident that many estimations can be considered almost the same within the (new) given error, so vanishing any possible inter-correlation and/or classification.

3. The two methods tend to behave differently within the range of the fractal dimension variation: for example the box-counting often tends to saturate when increasing the fractal dimension providing an under-estimation (please also look at Liang et al. 2012).

In addition to the above points that concern the way the methods are applied, I would like to add that there is some confusion when introducing some concepts like fractals, SOC and critical point, without state the most important differences among them.

Among the minor points, I do not understand the units of figure 2 (by the way, there is log r0 at the x-axis that I do not understand, too), that, strange enough, is given as example, but it is one of the two outliers of the overall analysis.

The last but not the least, the level of written English, which is very poor and plenty of refuses.

Therefore, I am sorry to say that my personal opinion is to reject the paper in the present form.

References

Liang et al., Comparison of fractal dimension calculation methods for channel bed pro-files, Procedia Engineering, 28, 252-257, 2012.
Theiler J. , Estimating fractal dimension, J. Opt. Soc. Am. A, 7, 6, 1055-1073, 1990.

---

## Author Comment (AC1) · 28 Dec 2019

dear Anonymous Referee,

Firstly I appreciate your comments and your time to read my manuscript. It also enlightens me to more comprehend what I am studying. The references Thelier (1990) gives more insight into the limitation of the method used. last, I give the responses below:

1. Too few data (especially in the aftershock spatial distribution) Author's Response: yes, it is because of the completeness of the catalog used.

[Figure]

2. Too small estimation of errors, because simply deduced from the linear regression in the log-log plot. If we estimate a more realistic error with an average of around three times that given, it is evident that many estimations can be considered almost the same within the (new) given error, so vanishing any possible inter-correlation and/or classification. Author's Response: yes, it is also the limitation of the method used in this study.

3. The two methods tend to behave differently within the range of the fractal dimension variation: for example, the box-counting often tends to saturate when increasing the fractal dimension providing an under-estimation (please also look at Liang et al. 2012). Author's Response: I used two different methods because of the two-point correlation integral, in my opinion, more accurate to estimate point (epicenter) distribution. But, the correlation integral can't estimate line (fault). So to calculate the fault I use box-counting.

In addition to the above points that concern the way the methods are applied, I would like to add that there is some confusion when introducing some concepts like fractals, SOC and critical point, without state the most important differences among them. Among the minor points, I do not understand the units of figure 2 (by the way, there is $\log r_0$ at the x-axis that I do not understand, too), that, strange enough, is given as an example, but it is one of the two outliers of the overall analysis. Author's Response: I'll correct the example and add the differences among fractals, SOC and critical point.

warm regard

---

## Author Comment (AC2) · 28 Dec 2019

Dear Francesco Visini,

first of all, I thank and appreciate your detailed comments, which of course take your time. the responses are in below:

Detailed comments. 23) Abstract: Please, introduce a brief explanation of fractal dimensions to make it easy to understand to a wide and diversified audience.

Author's Response: will be considered

Page 1 24) Line 18: Please specify what do you mean with "slip segmentation", it is

not clear.

Author's Response: it means that The Sumatran Plate Boundary is a slip-partitioned system. Dip-slip is accommodated across the subduction zone and strike-slip is accommodated by Sumatra Fault Zone (SFZ) as a dextral fault.

25) Line 20: I'd suggest to introduce here the figure1 with faults and seismicity to show the tectonics. Please, add also a wide tectonic scheme to introduce readers that are not-familiar with the region.

Author's Response: will be considered

26) Line 21-22: Please, add the references to support this.

Author's Response: The following references have been added: References : Bhattacharya, P. M., and Kayal, J. R.: Application of fractal in marine sciences: Study of the 2004 Sumatra earthquake (Mw 9.3) sequence in Andaman-Nicobar islands, Indian J. Mar. Sci., 36(2), 136–140, 2007. Roy, S., Ghosh, U. and Hazra, S.: Fractal dimension and b -value mapping in the Andaman-Sumatra subduction zone, Nat. Hazard, 27–37, doi:10.1007/s11069-010-9667-6, 2011. Sukmono, S., Zen, M. T., Kadir, W. G. A., Hendrajaya, L., Santoso, D. and Dubois, J.: Fractal Geometry of the Sumatra Active Fault System and Its Geodynamical Implications, J. Geodyn., 22(112), 1–9, 1996. Etc.

27) Line 25: the reference is Turcotte, 1997 (delete the name)

Author's Response: will be considered

28) Line 28: Please, add the references to support that GR is considered as SOC.

Author's Response: The following references have been added: References: Dimri, V. P.: Fractal Behaviour of the Earth System, edited by V. P. Dimri, Berlin., 2005. Donald L. Turcotte: Fractals and Chaos in Geology and Geophysics, 2nd ed., Cambridge University Press, New York., 1997.

Page 2 29) Line 4: I think it will be more clear than now if you remove ". In the same

study the correlation between both fractal values is SOC, the equation being as follows"
and just left ":".

Author's Response: will be considered

30) Line 8: please specify what do you mean with "satellite faults".

Author's Response: satellite faults are faults that surround the main-shock fault Yamashita and Knopoff (1987). Fir 1.

31) Line 13: "This result is more affected by the large uncertainty relative to the variation of D2.".Question1: More than what? Question2: is there a reference to support this statement? Or, if it is your idea, it has to be supported by a quantitative analysis.

Author's Response: it is from Padhy et al., (2013). Will be rephrased into "there is no correlation due to the large uncertainty of D2"

32) Line 22-25: Please rephrase because it is not clear.

Author's Response: The data were obtained from PuSGeN catalog (National Earthquake Study Centre) which was collected from International Seismological Centre (ISC), National Earthquake Information Centre U.S. Geological Survey (NEIC-USGS), EHB catalog (Engdahl et. al., 1998) and Badan Meteorologi, Klimatologi, dan Geofisika (BMKG) catalog that was relocated by Shiddiqi et. al. (2015).

33) Line 25: "This catalog has all the earthquake data with the magnitude above 4.5 Mw, starting from 1900 to 2016", Please add a reference or show a completeness analysis to support this statement.

Author's Response: Pusat Studi Gempa Nasional, Pusat Litbang Perumahan dan Permukiman.: Peta Sumber dan Bahaya Gempa Indonesia (in bahasa)., Pusat Penelitian dan dan pengembangan perumahan dan permukiman Kementerian Pekerjaan Umum dan Perumahan Rakya, Bandung, 2017.

Page 3 34) Line 2: "Active faults were identified based on morphology that had been

carried out in previous studies." Please add a reference

Author's Response: Pusat Studi Gempa Nasional, Pusat Litbang Perumahan dan Permukiman.: Peta Sumber dan Bahaya Gempa Indonesia (in bahasa)., Pusat Penelitian dan dan pengembangan perumahan dan permukiman Kementerian Pekerjaan Umum dan Perumahan Rakya, Bandung, 2017.

35) Line 3: "The SFZ was divided into 42 segments, while the WAF was divided into 4 segments" Who did this segmentation? and what it is based on?

Author's Response: Pusat Studi Gempa Nasional (PuSGeN) did it. It is based on morphology and earthquake distribution.

36) Line 4: Are there paleoseismological studies to support the activity of these faults?

Author's Response: yes, there are. I'd consider adding paleoseismology studies in the regional or tectonic setting in the manuscript.

37) Line 5: can you be sure this is not a bias due to the seismic network configuration? How did you select the aftershock? Is a time-radius magnitude dependent approach?

Author's Response: yes indeed, it is based on a time-radius magnitude approach.

Page 4 38) Regarding the mainshocks shown in the Table 1. You stated that "This catalogue has all the earthquake data with the magnitude above 4.5 Mw, starting from 1900 to 2016", but there are only earthquakes after the 1967 (I suppose this is the beginning of the instrumental period). So, this table is a selection of events? And if so, please specify the reasons. Or these are just the earthquakes with a correlation? And in this case you should show all the results.

Author's Response: the table contains selected events from the catalog. The event that has aftershock least than 9 earthquakes would generate high uncertainty (error).

39) At line 3 you wrote : "Tab. 1 is a tabulation of the results of the identified aftershocks and the correlation result". It is not clear if Tab 1 list the mainshocks. Please, specify

how you build this table.

Author's Response: Tab. 1 means earthquake event that was identified or selected, including aftershocks and main-shocks.

40) Finally, please use the dot for decimals.

Author's Response: will be considered

41) Eq2 and later: Please use the same letter for distance R or r. Or are them different?

Author's Response: they are different in that eq. R means distances between epicenters and r mean given distance that was used in the calculation.

42) Line 11-13: Please rephrase because, I think a verb is missing. It is not clear here, how D2 was estimated. Please consider to explain variables in the order they are presented.

Author's Response: D2 was estimated by $(r) \sim r\hat{}(D\_2)$, and to calculate it I used the least-square fit method. C(r) is the correlation of integral and r means given distance that was used in the calculation.

43) Line 6: "Conversely, a small D2 value indicates a tight earthquake distribution.", please specify what do you mean with a tight earthquake distribution. It is not clear if you are referring to the epicentral distribution or something else.

Author's Response: it means dense epicentral spatial distribution.

44) Line 6: delete "Donald L. "

Author's Response: will be considered

45) Line 7: The box-counting method is to make boxes in the aftershock cluster, please rephrase, because it is not clear.

Author's Response: In this method, square grids that cover the aftershock cluster were created.

46) Line 9: "length of the box (r)". This is the first time you mentioned length of box. Define all the parameters you are using to do your calculations.

Author's Response: yes, I would redefine the parameters

Page 5 47) Line 6: please correct in: "Figure 2 shows". Specify that this is an example.

Author's Response: yes, I would

48) The C(r) value calculation was conducted by using Python 3.7 Python is a language, what do you mean? Is there a specific function that you used?

Author's Response: I'd consider deleting this sentence. Functions used are R =cos−1 {cosðİŽij1cosðİŽij2+ sinðİŽij1sinðİŽij2cos(ðİŽ¡1−ðİŽ¡2)} and ðİŘ𝗎́(ðİŚ§)= 1ðİŚĄ2Σ𝗈ðİŘ𝗓(ðİŚ§−ðİŚĚ).

49) D2 estimation has a fairly high level of biases due a lack of complete catalogue or limited number of data points (Padhy et al., 2013). Can you give a quantitative analysis of this?

Author's Response: it was mentioned in Padhy et al., 2013.

50) Line 8-9: "The estimations were performed multiple times in several points to find a lower error value as done by Nanjo & Nagahama (2004)", please specify lower than what.

Author's Response: the lowest.

51) Here you are introducing "points" for the first time. What are them? They are numbered, but you do not specify what is the range. Please specify what these points are and, if possible, add them in figures.

Author's Response: the points are the projection points in r and C(r) curve or r and N(r) curve, which were the result of the calculation. I would add them in figures.

Page 6 52) Line 10: "We found three correlations and two earthquake occurrences that

did not follow the three correlations (Tab. 1 and Fig. 4)", do you mean that "Basing on values of slopes and intercepts of Do versus D2, we identified 3 groups of D2-D0. For the first group, we calculated a correlation with a slope greater or equal than 1.6 and an intercept of X.X. For the second group, we calculated a correlation with a slope greater or equal than xxx and an intercept of X.X.….? Please, rephrase this sentence because it is not very clear.

Author's Response: will be considered

Page 7 53) Line 7: However, that study was outside of our scope. What do you mean?

Author's Response: Bhattacharya and Kayal (2007)'s study was outside our study area.

54) Line 8: "The results obtained are relatively similar, but there is a significance difference in Oreng and Aceh-Center segment. It is due to differences in the morphological interpretation of active faults in that segments. Please, discuss in detail what are the reasons for such a differences, and how you define if they are significant or not.

Author's Response: the reason is the difference in active fault interpretation (completeness of fault database) between this study and the study from Sukmono et al., (1996). In my study, significant means that there is a 0.1 difference in D0 between both studies.

55) Line 13: "The third and the first correlation show a nearly identical value with only 0.06 differences". Please specify which value, I think it is the slope, but you should be clear.

Author's Response: yes, it is the slope. I'd be more specific.

56) Line 13: " A significant difference in the intercept shows the scale variation". What do you mean with scale variation?

Author's Response: it means scale invariance.

57) Line 15: We estimated b-value variation for both correlation 1 and 3 to find out whether there was an influence of the value (Tab. 2). How do you calculate the b? in

table 2 you show b value much higher than 1-1.2, this is quite strange and needs to be explained in detail. Is that the b- of the sequence? Please specify in detail your computation

Author's Response: to calculate b-value I used a least-square fit method. It is the b-value of aftershock and main-shock sequences.

58) Line 18: "Different catalogue also did not show significant differences." For example, b-value moves from to 2 to 1.3, is that not significant?

Author's Response: I mean the significant difference between correlation 1 and 3. I would rephrase the sentence.

59) Line 18-21: The number of the aftershock looked quite significant. In correlation 1, the number of aftershocks was above 18, whereas in the third the opposite occurred. However this result also cannot strongly explained why there was a scale difference in the number of aftershocks generated. Please rephrase this sentence because it is not clear. What do you mean with "opposite" and "aftershocks generated" ?

Author's Response: opposite mean in the correlation 3 the number aftershocks below 18. "aftershocks generated", I would delete the "generated" word.

Page 8 60) Line 10: correct in 0.4

Author's Response: yes, thank you

61) I'd suggest to add a column in the table with the difference in the occurrence between mainshock and largest aftershock (in years).

Author's Response: yes, I would

Page 9 62) Line 5-9. This paragraph should be completely re-written, it is not clear.

Author's Response: the earthquakes occurred because of the satellite fault in the near of main-shock cluster. The main-shock of Nicobar 2015 earthquake might be the Nico-

bar 2014 earthquake and the Tapanuli 2005 might be the satellite fault from the main fault of Mentawai 2005.

63) In the discussions, there are no clues about the possible effect of completeness of the fault database and earthquake catalog on estimations of D2 ad Do and their correlation. Do the correlations show a spatial significance or not?

Author's Response: yes, indeed. The completeness of the fault database and earthquake catalog would be significant. I would discuss the effect in the revised version. The correlation shows a spatial difference but I have not discussed it in this manuscript because I need further research.

64) The last conclusion, "Finally, this study can be used as a model to predict the spatial distribution of aftershocks with variations of general earthquakes, doublet earthquakes and scales", it is not discussed in the manuscript before. If number of aftershock impact the bvalue, and the correlations are done on basing on the observed number of aftershock, how is it possible predict spatial distributions? What do you mean with general earthquakes? Scales? Please, discuss this point in the discussion before state it in the conclusion.

Author's Response: I would discuss it first in the discussion section in the revised version. It can only predict the density of aftershock spatial distribution (D2) from the distribution of active fault. General earthquake means the earthquake that was caused by general mechanism. This term was used to differs the afore-mentioned earthquake to doublet earthquake that has a different mechanism Scale means scale invariance of the active fault.

best regard.

———————————————

[Figure]

**Figure 2.** Schematic diagram for Model II. The shaded area M denotes the main-shock slip zone. The smaller cracks $S_i$, denote small satellite faults that surround the main-shock fault. The gaps between the satellites and the edge of the main shock fracture disappear by stress corrosion.

**Fig. 1.**